# Thermo-Catalytic Decomposition Comparisons: Carbon Catalyst Structure, Hydrocarbon Feed and Regeneration

Mpila Makiesse Nkiawete and Randy Vander Wal *

The EMS Energy Institute and the Department of Energy and Mineral Engineering, The Pennsylvania State University, University Park, PA 16802, USA; mmn19@psu.edu
* Correspondence: ruv12@psu.edu

**Abstract:** Thermo-catalytic decomposition (TCD) activity and stability depend upon the initial carbon catalyst structure. However, further transitions in the carbon structure depend on the carbon material (structure and composition) originating from the TCD process. In this article, reaction data are presented that illustrates the time-dependent TCD activity as TCD-formed carbon contributes and then dominates conversion. A variety of initial carbon catalysts are compared, including sugar char, a conductive carbon black (AkzoNobel Ketjenblack), a rubber-grade carbon black (Cabot R250), and its graphitized analogue as formed and partially oxidized. Regeneration of carbon catalysts by partial oxidation is evaluated using nascent carbon black as a model, coupled with subsequent comparative TCD performance relative to the nascent, non-oxidized carbon black. Activation energies for TCD with nascent and oxidized carbons are evaluated by a leading-edge analysis method applied to TCD. Given the correlation between nanostructure and active sites, two additional carbons, engine soots, are evaluated for regeneration and dependence upon nanostructure. Active sites are quantified by oxygen chemisorption, followed by X-ray photoelectron spectroscopy (XPS). The structure of carbon catalysts is assessed pre- and post-TCD by high-resolution transmission electron microscopy (HRTEM). Last, energy dispersive X-ray analysis mapping (EDS) is carried out for its potential to visualize oxygen chemisorption.

**Keywords:** thermo-catalytic decomposition; nanostructure; active sites; natural gas; HRTEM

## 1. Introduction

As an energy carrier, hydrogen is the basis for the proposed hydrogen economy. As a fuel, hydrogen is versatile, being suitable for use in internal combustion engines, gas turbines, heat generation, or fuel cells for generating electricity without greenhouse gas emissions [1,2]. As a chemical, hydrogen is key to converting $CO_2$ emissions into platform chemicals. Hydrogen has direct manufacturing utility for steel. Ultimately, hydrogen could serve as a sustainable renewable energy resource, buffering fluctuations [3,4]. Presently, a significant amount of hydrogen is produced from upgrading fossil fuels, mainly petroleum, into various liquid fuels for transportation.

Today, the majority of the hydrogen is produced by steam methane reforming (SMR). Although SMR is more energy efficient than TCD, once carbon capture is included, efficiency drops. While SMR has high energy efficiency (~75%), the penalties imposed by carbon capture (~30%) and storage (~30%) together result in a substantial drop in energy efficiency [5]. Moreover, the high electricity consumption from existing sources increases the environmental impact [6]. Based on thermodynamic enthalpies and including the energy input for water-gas shift (WGS), SRM requires 63.4 kJ for one mole of hydrogen, compared to 37.4 kJ for TCD [7].

Thermo-catalytic decomposition (TCD) provides a way to decarbonize natural gas, producing only solid carbon as a byproduct. Advantageously, no WGS stages or CO removal are required. Though long studied for its energy costs, with pilot-scale demonstrations

in operation, $CO_2$ capture and sequestration impose substantial capital and operational capture costs [8], often not considering downstream delivery infrastructure [9], and with few, if any, storage options in most cases [10]. In contrast, TCD provides a potential bridge to the hydrogen economy while using abundant natural gas resources and existing production and delivery infrastructure. While solid carbon can have commercial value in manufactured products [11], substituting for carbon blacks, activated carbon, etc., at large scale, other applications such as environmental remediation and soil enrichment become tenable [12], and in fact, it is necessary if at very large scale [13].

While the majority of TCD studies have focused on metal catalysts, the process is then akin to carbon nanotube synthesis or filament formation, with the eventual outcome of catalyst deactivation by coking [14,15]. Though a valued allotrope, all the residual issues pertaining to carbon nanotube (CNT) synthesis arise, including CNT harvesting and catalyst removal, generally requiring a combination of oxidation and acid etching, both of which degrade the CNTs, decrease yield, and impose environmental issues related to disposal [16,17].

Carbon materials as catalysts are generally less active than metals, requiring higher temperatures (800–1000 °C). Yet carbon catalysts are more stable at high temperatures, i.e., offering a longer lifetime with feed flexibility while being relatively immune to poisoning [13,18]. Moreover, facilitated by material similarity and low cost, they can become part of the TCD carbon, avoiding any subsequent recovery or removal processes [13,19]. In carbon materials, there is no singular definition of active sites; they are often described as including lattice dislocations, low-coordination sites, vacancies, discontinuities, edges, defects, and other energetic abnormalities [20,21]. Thus, more disordered carbons should be more active than highly ordered carbons [22,23]. Supporting this view, activated carbons have exhibited greater stability than carbon blacks, but porous carbons have a finite capacity, defined by their porosity.

Literally dozens of carbons have been tested, including graphite, diamond powder, carbon nanotubes, glassy carbon, fullerene soot, fullerenes C60/70, acetylene black, coal char, and ordered mesoporous carbons (CMK materials) [13,19,24]. A less ordered structure is generally recognized as advantageous because it hosts more high-energy sites (i.e., active sites). Yet comparisons of different carbon structures are few, with nearly no measurements of active sites pre- or post-TCD or after regeneration, though TCD rates are contingent upon their concentration and TCD stability (long-term activity) dependent upon their continuity or continuing recreation.

Long-term, carbon materials deactivate, though most tend to settle to some steady state, albeit with low activity—carbon blacks in particular [13,19,25,26]. Oppositely porous carbons exhibit a greater level of deactivation, reflecting their pore-based capacity for carbon deposition [13,19]. As a means of retaining activity, various hydrocarbons have been co-fed with methane to retain activity [13,19,27]. Maintenance or increase in catalytic activity is attributed to smaller crystallite sizes associated with the deposit produced by the mixed gas feed, fostering a higher surface concentration of active sites. A significant point to consider in these additive studies, let alone nearly all TCD studies for rates, activation energies, carbon catalyst comparisons, etc., is that laboratory-grade (pure) methane is used, presumably greater than 98% purity [13,19,28]. Yet TCD deployed at any scale will utilize pipeline natural gas, for which there is only a compositional range defined by industry, but most notably, not pure methane. Notably, for field applications, in lieu of flaring, unprocessed (raw) natural gas would be processed. Table 1 lists the components and compositional ranges for natural gas in the US for reference. A second key point is that most TCD studies evaluate kinetics while referencing stability and activity for the initial catalyst [13,19,28,29]. Though structural analysis and other characterizations are straightforward, nearly all initial carbon catalysts are formed under conditions very different from TCD carbon. To mention a few, ordered mesoporous carbons are formed by templating followed by high temperature carbonization [30,31], activated carbons are formed from high oxygen content biomass followed by physical or chemical activation [32,33], and car-

bon blacks are formed from refinery distillation residual or similar byproduct feedstocks such as ethylene tar in high temperature reactors approaching 1800 K [34,35]. Therein lies the importance of evaluating TCD with TCD carbon. Ironically, TCD is the very definition of catalyst "coking." Once porous carbons are filled or non-porous carbons are filled, the deposited TCD carbon will soon govern subsequent conversion with stability dependent upon the continuity of active site formation by continuing carbon deposition—driven by structure; which in turn is determined by the process temperature, hydrocarbon feed, and its pyrolysis products within the TCD reactor.

**Table 1.** Natural Gas Composition Range.

| Natural Gas | Component | | | |
|---|---|---|---|---|
| Mole (%) | Methane | Ethane | Propane | Iso-Butane |
| Typical Analysis | 94.7 | 4.2 | 0.2 | 0.02 |
| Range | 87.0–98.0 | 1.5–9.0 | 0.1–1.5 | Trace–0.3 |

To address some of these gaps in knowledge, this study compares different initial carbons as catalysts, resolving initial versus longer-term deposition, with the latter connected to the deposited carbon structure, examined by HRTEM post-TCD. The different carbon materials afford the opportunity to compare the structure of the TCD carbon and that of the nascent carbon catalyst. The potential of regeneration for active site formation is evaluated with a model carbon black, while two other carbons, engine soots, are evaluated after mild oxidation by HRTEM owing to their distinctly different nanostructures.

The present manuscript conveys several points of novelty.

1.  Using a set of model non-porous carbons, a comprehensive set of comparisons is made for reaction metrics: onset temperature, peak conversion, and long-term activity/stability.
2.  A realistic, multi-component synthetic natural gas is used at 100%, as opposed to pure laboratory-grade methane. Using the same carbon catalyst, we find significant differences in reaction metrics and TCD carbon structure.

In particular, this study shows significant differences relative to pure methane for reaction metrics and TCD carbon structure using a generic, non-porous carbon black. (See results Sections 2.1 and 2.2, contrasting R250 with pure methane versus synthetic natural gas, respectively).

3.  Carbon catalyst nanostructure dependence for reaction metrics (nascent and graphitized carbon black) with the same catalyst particle size, morphology, surface area, and absence of porosity is demonstrated.
4.  A new methodology for active site measurements is presented: XPS following activated chemisorption, featuring direct quantification of chemisorbed oxygen with potential differentiation of carbon-oxygen functional groups.
5.  The first intensive study of TCD carbon structure and morphology by HRTEM.

## 2. Results

### 2.1. Nanostructure Comparisons: Sugar Char versus Carbon Black

Given the mixed contributions from the initial and depositing carbon, two regimes are of particular interest. The first is where the initial catalyst dominates, governing the observed rate. Here, direct dependence of rate upon nanostructure can be made by direct comparisons across different carbon nanostructures using methane.

The first comparison is between char and conductive carbon black. The TCD curves for each appear in Figure 1. Onset temperatures for the char are different, approximately 650 °C for the char and 750 °C for the carbon black. Peak conversions are also quite different. ~33% versus ~52% for the char and carbon black, respectively. Most significant is

the recovery to asymptotic levels where the TCD carbon is governing deposition. The char is minimally active, ~3% at long durations, while the carbon-black-supported TCD carbon maintains a relative activity of 30%.

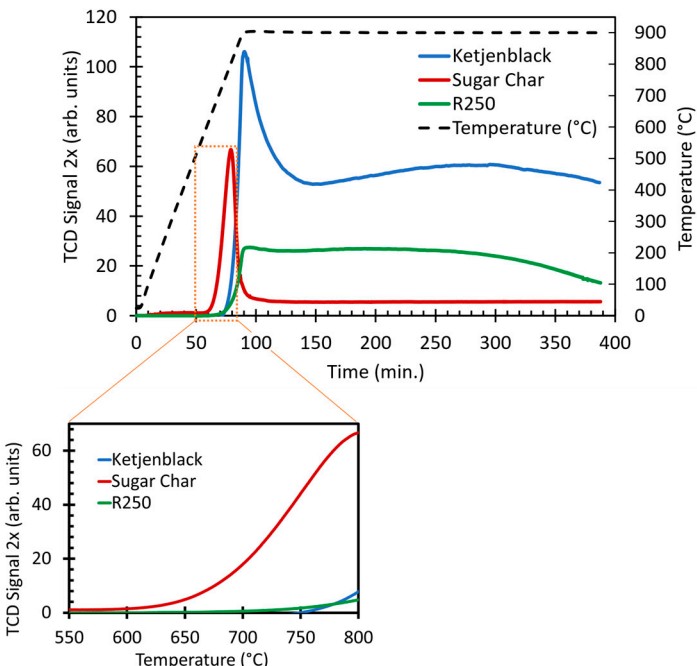

**Figure 1.** TCD conversion rates for char and conductive carbon black as a function of time. The temperature ramp is illustrated by the dashed curve, plotted on the 2nd *y*-axis.

By XPS analysis, chemisorbed oxygen content as a measure of active sites was less than 1 at.% for the conductive black and R250, while 3.4 at.% for the char. While the initial onsets and rates correlate with active sites, the activity of the TCD carbon anti-correlates, demonstrating a complex relationship between deposit nanostructure and that of the nascent carbon.

These different behaviors are related to nanostructure. At short reaction times, the conversion is due to the initial carbon activity, but at long durations, it depends on the TCD carbon activity. From Figure 1, the different onsets and peak conversions are attributed to the different lamellae nanostructures between the char and carbon black.

As the TEM images in Figure 2 show, the nascent sugar char is nearly amorphous, while the conductive black presents extended lamellae with proportionally fewer edge sites. One might expect an amorphous carbon with aromatic clusters for lamellae to host more active sites than a carbon formed by high-temperature pyrolysis (specifically acetylene pyrolysis for the conductive black). However, the opposite appears to be the case. Formed at a low temperature of 600 °C, the early char behavior is attributed to annealing in the reducing TCD environment. The annealing is apparent by the visible lattice fringes forming shells, capsules, and stacks in the sugar char, as found post-TCD. Either this annealing provides fewer TCD growth sites and/or the resulting TCD carbon is less active than that on the conductive carbon, which does not exhibit any observable change in structure after TCD compared to its nascent form. The fact that the TCD carbon on the sugar char is less active than that on the conductive black seems unlikely given the similarity of coral-like graphene stacks that appear on both carbons post-TCD. While the initial onsets and rates correlate with active sites, the activity of the TCD carbon anti-correlates, demonstrating a complex relationship between deposit nanostructure and that of the nascent carbon.

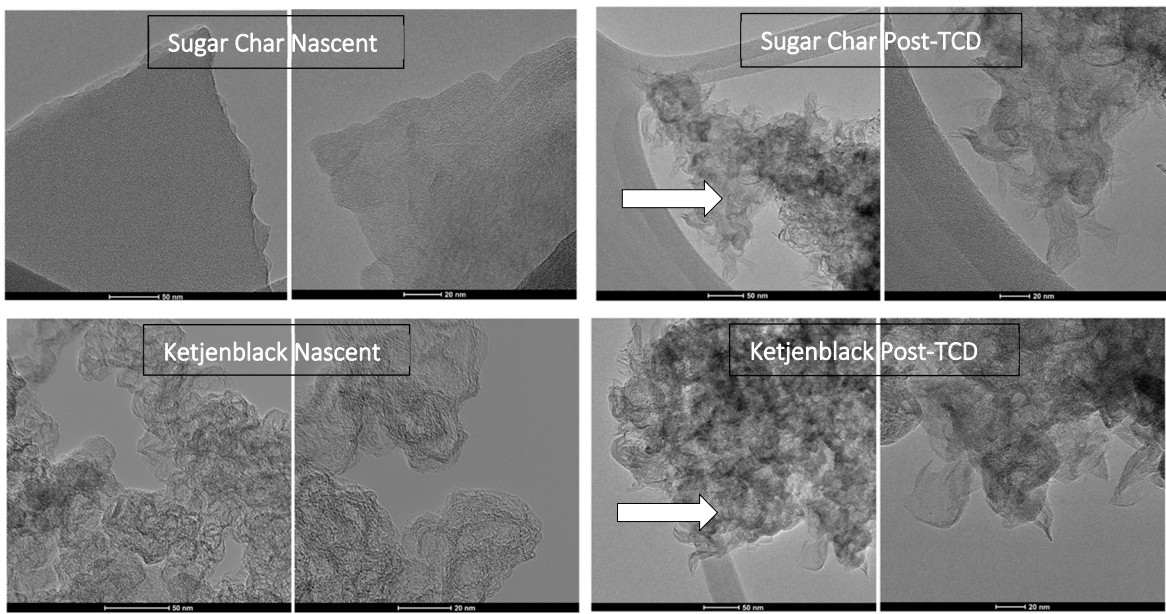

**Figure 2.** TEM images of the sugar char and the Ketjen carbon black pre- and post-TCD. Both carbons show graphene outgrowths, resembling coral.

## 2.2. Kinetic Rates and Nanostructure: Pre- and Post-TCD

To compare reactive sites, a comparison study utilized the R250 and its graphitized form, heat-treated at 3000 °C in a commercial induction furnace. As shown previously, the graphitized R250 (G-R250) exhibits a rhombohedral form, with sharply angled corners delimiting the length of the lamellae stack comprising a side of the particle. A typical wall is comprised of 10 or more parallel, uniformly straight lamellae, as seen in Figure 3.

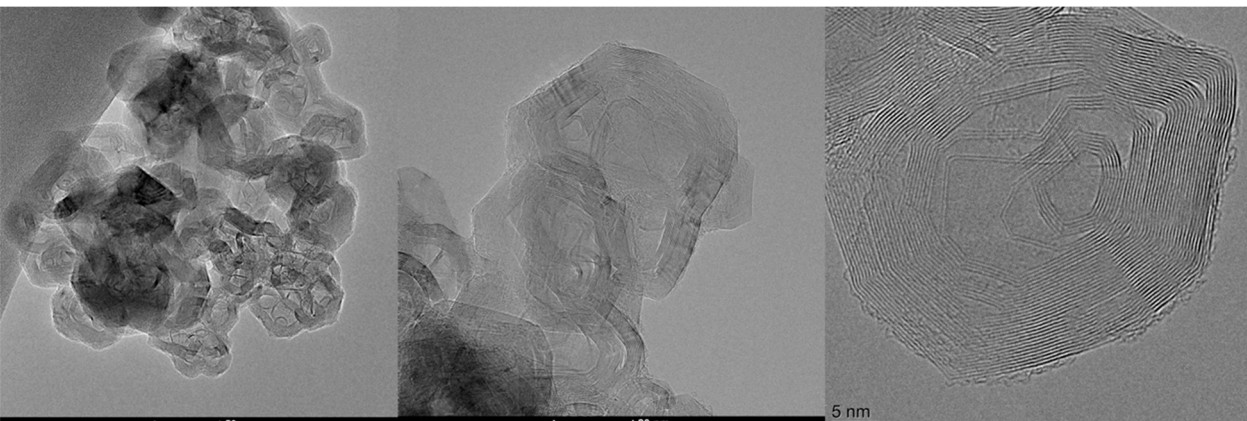

**Figure 3.** TEM images of graphitized R250 at selected magnifications.

The operative hypothesis was that the G-R250, featuring solely basal plane carbon, would offer no active sites to initiate carbon deposition. Therein, it would be comparatively inert and serve as a baseline by which to gauge not only the non-graphitized form of R250 but all such carbon blacks exhibiting lamellae segments along their periphery. Yet as the comparative rate plot in Figure 4 shows, the "inert" G-R250 exhibited an earlier onset temperature, a faster initial rate, greater peak conversion, and, as the final surprise, a higher level of conversion at long reaction times. TEM analysis of the G-R250 revealed a contrasting structure relative to the nascent R250, but not as expected. Figure 5 shows comparative TEM images for the nascent R250 and G-R250 forms. The nascent form appears as expected, with lamellae definition increasing towards the particle perimeter, culminating with short, poorly stacked lamellae defining the particle periphery. While

the G-R250 interior retains the afore-described graphitic structure, the particle exteriors appear to consist of a layer of poorly structured carbon, as marked by the arrows. This unstructured carbon coating appeared on all the particles with a similar thickness of a couple nanometers. What lamellae were recognizable appeared to be short, wrinkled, and lacking stacking. Such an amorphous carbon would be expected to host numerous active sites, and with its pervasive presence, this accounts for the kinetic observations. Indeed, the earlier onset temperature suggests that the activation energy for the G-R250 form is significantly lower than that for the R250 (166.5 kJ/mol and 190.0 kJ/mol for G-R250 and R250, respectively).

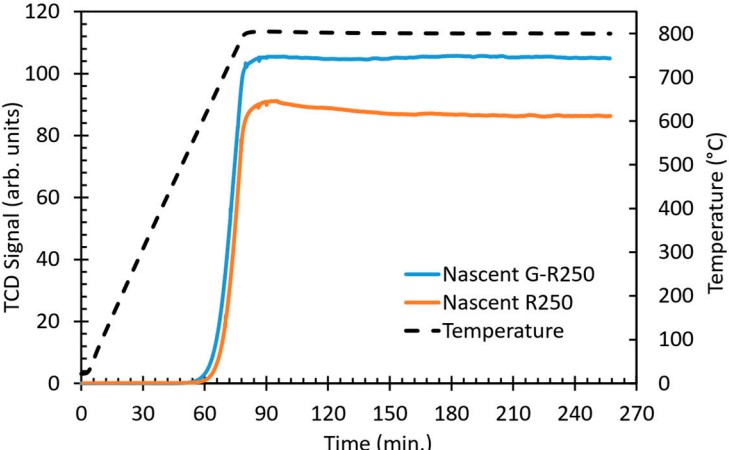

**Figure 4.** Comparative TCD rates for nascent R250 and G-R250; both run using synthetic natural gas at 800 °C.

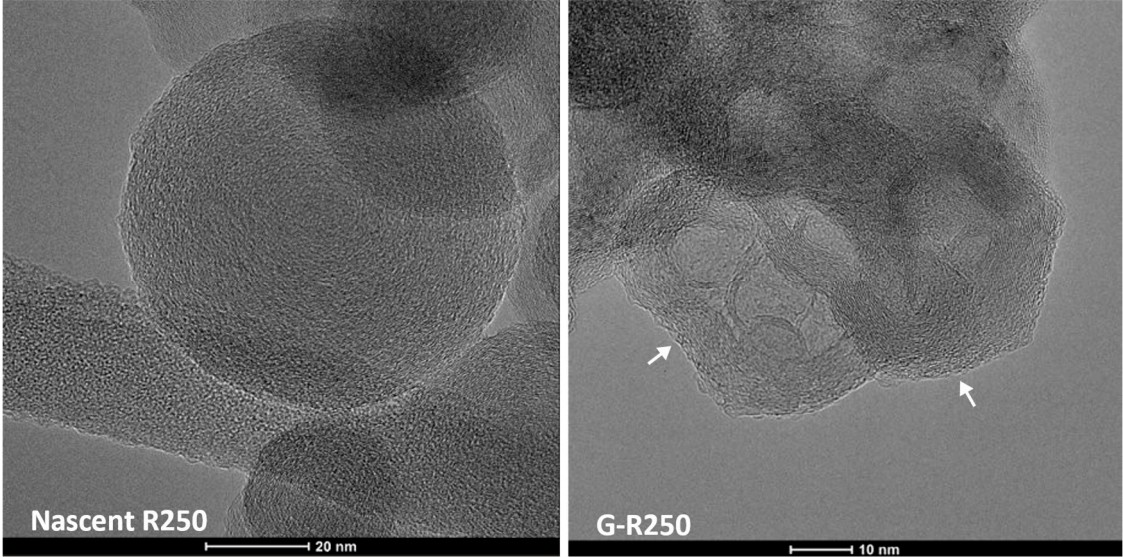

**Figure 5.** TEM images for the nascent R250 and G-250 forms. The former exhibits short lamellae with low stacking, while the G-R250 form has highly parallel and stacked lamellae yet has an unstructured carbon coating on the particle exteriors (indicated by the arrows).

The higher long-term activity of the G-R250 form likely reflects the contribution of the TCD carbon, but importantly, the TCD carbon activity appears to depend upon the nanostructure of the nascent catalyst. For both runs shown in Figure 4, the reactant, synthetic natural gas mixture (SNG), temperature, and space velocity are the same. The difference is between the nascent carbon catalyst and the exposed nanostructure. It would

appear that the unstructured carbon coating on the nascent G-R250 leads to a more active TCD carbon.

To further investigate this conjecture, TEM was performed on the G-R250 catalyst post-TCD. As seen in Figure 6, nascent R250 with SNG does not show the carbon coral as found for the nascent R250 carbon with pure methane. Instead, carbon particle structures often appear stacked in linear and angled arrays, somewhat analogous to carbon black but not clearly in the form of aggregates. Generally, the TCD (particle-like) carbon appears less structured than the nascent R250. In other locations, the TCD carbon appears to have connected with the edges of short lamellae (presumably the restructured amorphous carbon), as marked by the arrow in Figure 6. Numerous examples of this interfacial extension of lamellae were observed, suggesting a form of templating wherein depositing carbon supplied the feed for lateral lamellae extension. Notably, this templated carbon using SNG differs from the carbon coral observed on the sugar char (Figure 2) using pure methane. In contrast to the nascent R250, the G-R250 does support a coral-like form of TCD carbon. Irregular sheets with partial folds and amorphous wraps are observed. The greater edge site exposure of this TCD carbon form is consistent with its higher activity, as illustrated in Figure 3. Such contrasting image data confirms that the TCD carbon nanostructure does depend on that of the nascent carbon catalyst. Perhaps more importantly, the activity of the TCD carbon depends upon its nanostructure, and hence, so too do the active sites supported by the nanostructure.

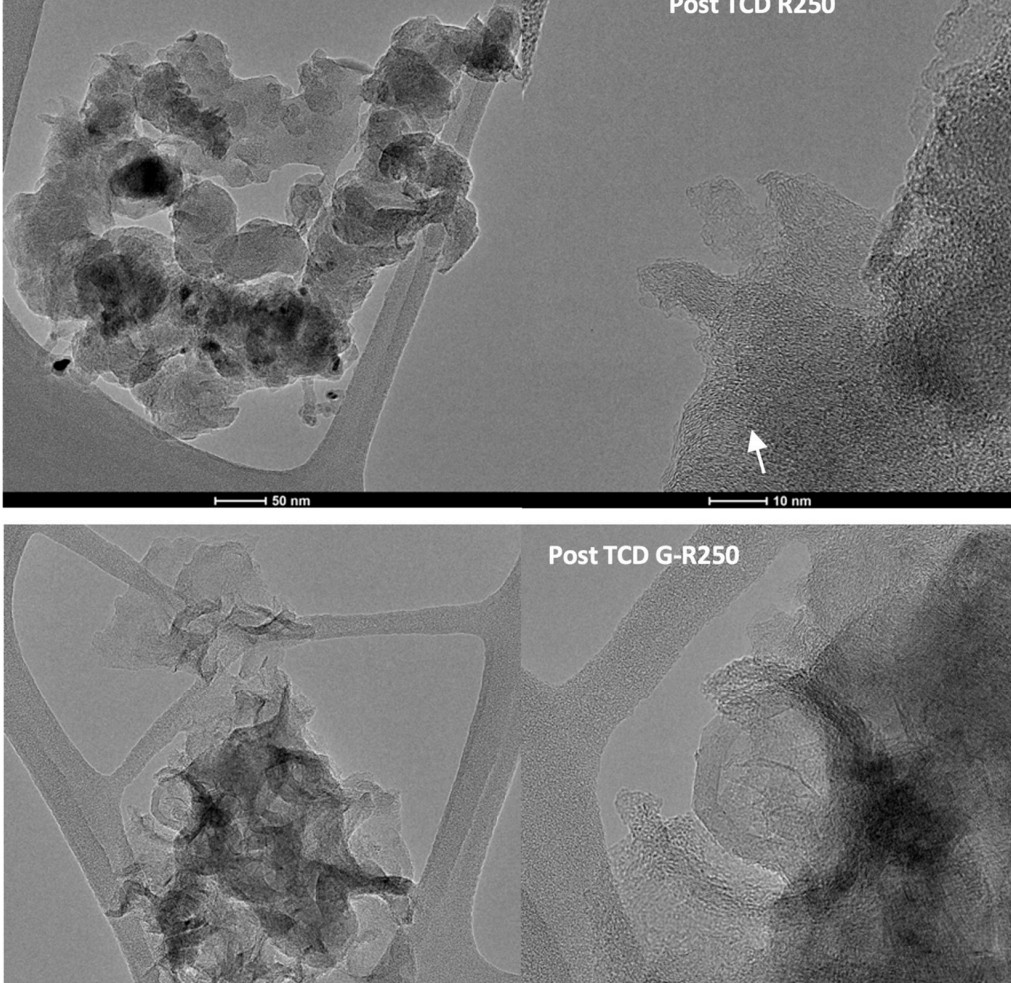

**Figure 6.** Contrasting TEM images for R250 and G-250, both post-TCD.

### 2.3. Origin of Unstructured Carbon Coating

Although the experimental results are consistent with the nanostructures, the underlying question is the origin of the amorphous carbon coating. Several prior publications presenting TEM images of the G-R250 form showed clean basal planes with negligible carbon debris, as illustrated in Figure 3. Our current speculation is that even though graphite is the most stable form of carbon, the rhombohedral form of the R-250 particle is highly stressed due to the sharply angled corners. This stress facilitates the oxidative breakup of exposed lamellae, resulting in the amorphization of the outer lamellae over time. The breakup would account for the numerous curled graphene segments seeking edge-site termination to minimize their free energy. That such oxidative breakup of seemingly stable "graphitic" lamellae occurs at ambient temperature suggests that the strain on these lamellae is considerable. It should also be noted that the G-R250 carbon was formed at 3000 °C, and the thermal contraction associated with an 11-fold decrease in absolute temperature could result in substantial stress as well.

### 2.4. Partial Oxidation for Deactivation

Given the difference in nanostructures between the amorphous carbon coating and the underlying graphitic carbon, partial oxidation was employed to preferentially remove the exterior carbon by exploiting the difference in oxidative reactivity. With an estimate of the amorphous content as less than 10%, partial oxidation of the nascent G-250 was performed in situ to a 10% burn-off level. Then, TCD was performed with the same conditions as for the nascent G-R250 carbon; the respective TCD curves are shown in Figure 7.

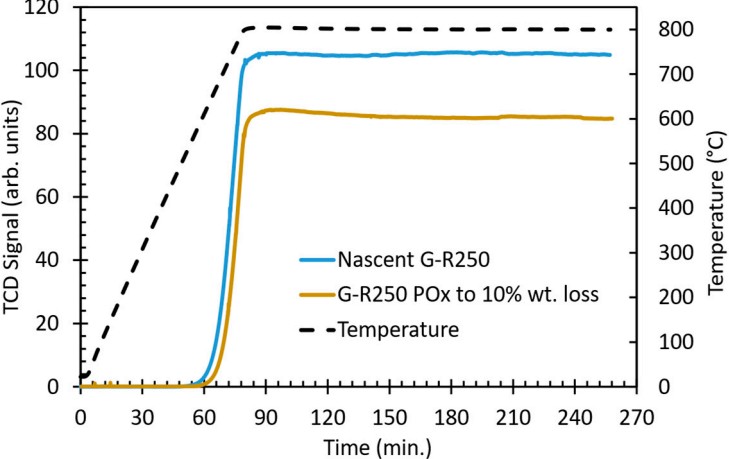

**Figure 7.** TCD runs utilizing nascent G-R250 and its partially oxidized form, with SNG at 800 °C.

As seen in Figure 7, the partially oxidized, "cleaned" G-R250 carbon has a higher onset temperature, lower peak conversion, and lower average conversion at long reaction durations where the TCD carbon is governing activity. This difference in TCD activity over long periods, wherein the TCD carbon is governing the activity again, suggests a dependence of TCD nanostructure and active sites upon the nascent carbon, as noted previously.

Based on these TCD results, it could be surmised that the partial oxidation did remove the unstructured carbon coating, leaving a less reactive carbon, exposing solely basal planes. Yet if that were the case, then what accounts for its relatively high remaining activity? Indeed, the onset temperature, peak, and asymptotic conversion levels are similar to the nascent R250 (non-graphitized) and G-R250 carbons by comparison to their respective TCD data as presented in Figure 4. However, the result of partial oxidation is less clear upon TEM examination. The partial oxidation, performed at a low temperature of 600 °C did not remove the unstructured carbon coating as presumed but instead amorphized the unstructured carbon coating while apparently increasing its thickness, as illustrated by the

arrows in Figure 8. This amorphous carbon clearly retains significant activity, as seen with reference to the nascent G-R250 (Figure 7) and nascent R250 (Figure 4). Lastly, it is important to note that, with the intent of removing unstructured carbon, this partial oxidation step was intended to deactivate the carbon. Normally, partial oxidation is intended to regenerate the carbon by creating new active sites. That comparison is presented in Section 2.7.

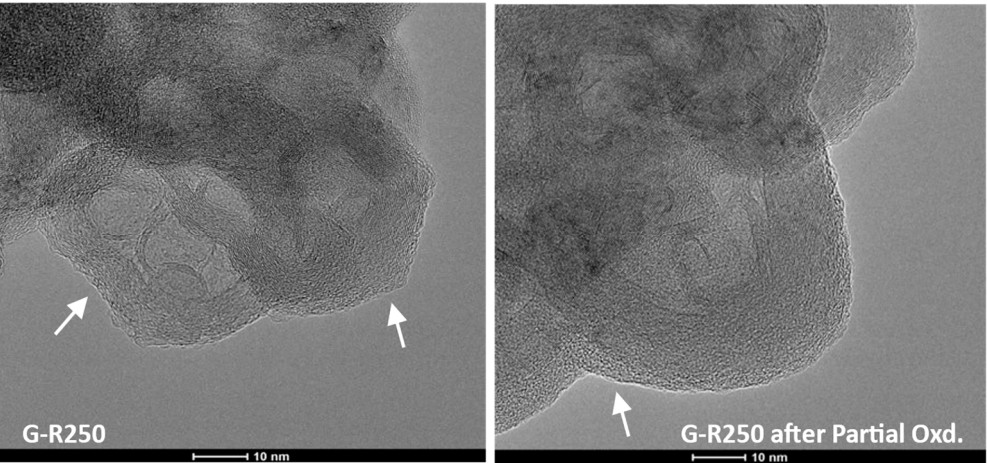

**Figure 8.** TEM images contrasting G-R250 nascent and after partial oxidation (10% burn off). Both images are prior to TCD testing.

### 2.5. TCD Carbon Nanostructure

To disentangle the TCD carbon structure dependence upon nascent carbon and hydro-carbon feed, TCD was conducted using R250, but now with methane with HRTEM applied post-TCD.

Figure 9 shows TEM images of the carbon black (R250), nascent and recovered from the packed bed after TCD using methane. The nascent form has visible lamellae configured in the usual concentric manner about the particle core with decreasing structure towards the particle interior. Most notably, the particle surface is comparatively "smooth" relative to the recovered catalyst, as seen in Figure 9. Outgrowths and protrusions of coral-like carbon are observed, in contrast to the particle morphology found with R250 using SNG.

The formation of ill-structured TCD carbon upon initial carbon catalysts is surprising. As a pure pyrolysis process without external forces such as electric or magnetic fields, the formation of semi-layered lamellae would be expected, certainly with the precedence of pyrolytic carbon and soot growth in combustion environments. The carbons have no metal contaminants, nor were any particles detected by TEM. Given that similar TCD carbon forms on both the sugar char and Ketjen carbon black, this further implies that the TCD carbon structure is independent of the nascent carbon catalyst structure; rather, it is driven by the pure methane feed. However, the lateral concentration or "density" of such outgrowths may be dependent upon the concentration of active sites and possibly their grouping that support the initial carbon addition reactions. Further investigations are warranted. Neither the char nor carbon black are porous; their surface area is solely external. A particular advantage of the carbon blacks is their well-defined particle structure and rather uniform size—providing a clear interface by which to distinguish any deposited carbon—and none was observed. All the deposited carbon appeared to be in the form of corals. Additional questions raised by these observations include the prevalence of such carbon coral and whether it alone accounts for the observed TCD activity at long reaction times. If truly dependent upon feed and, as observed here, specific to laboratory-grade methane, then TCD conversion kinetics reported in the literature are highly specific to this deposit structure and methane, making extrapolation to real natural gas problematic.

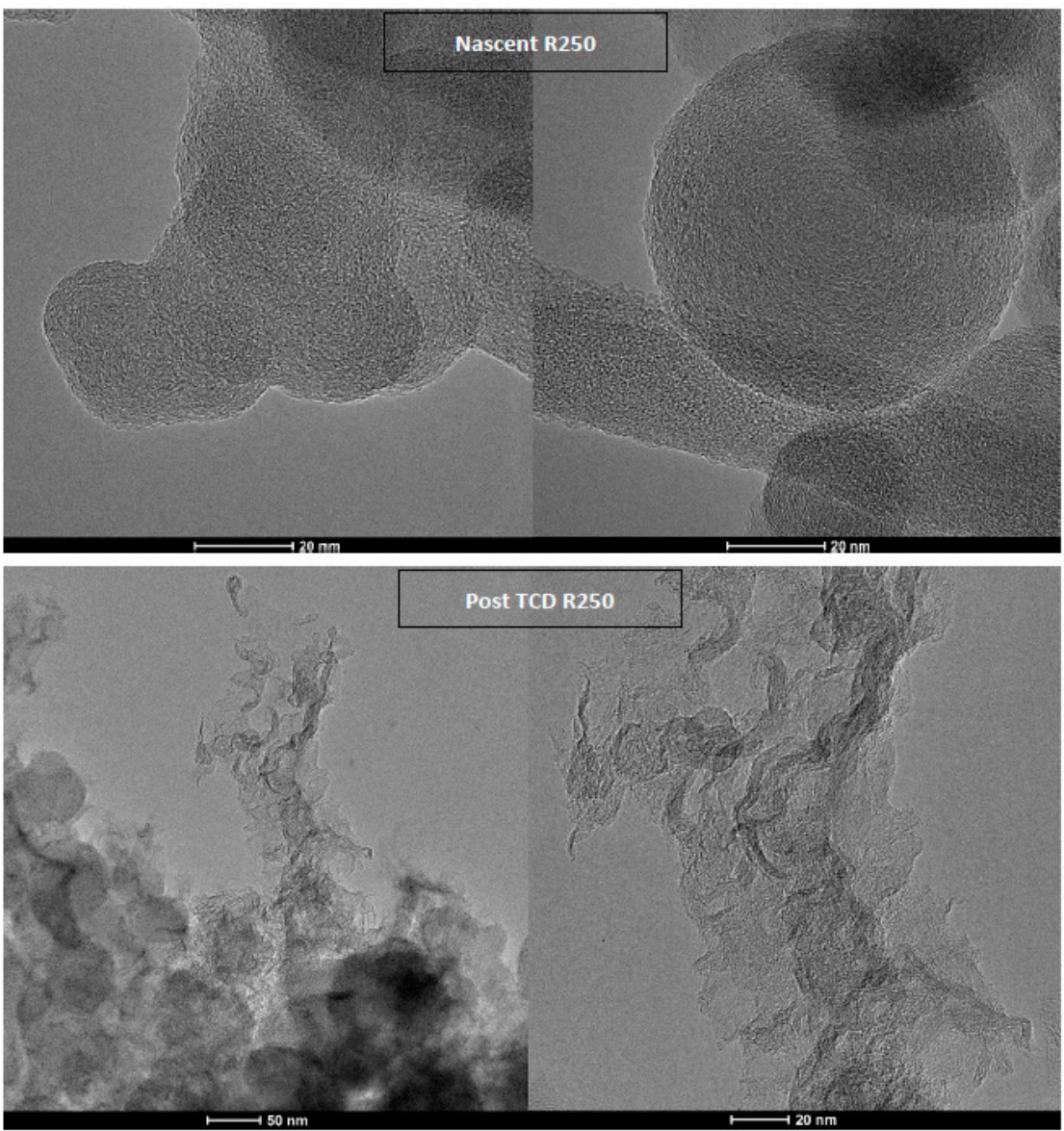

**Figure 9.** TEM images of nascent and post-TCD carbon catalysts (R250), the latter showing "carbon coral".

The observed carbon coral also generates questions regarding the deactivation observations as reported here and observed in other studies. Filling of pores or loss of surface area and active sites are physically intuitive explanations. The latter corresponds to the formation of layered lamellae, akin to soot particle growth. Outward-directed growth into 3D disallows this conceptual model and explanation. As noted in our review, TEM of carbon catalysts post-TCD is nearly absent [36]. Moreover, handling and sampling issues may arise, e.g., location in the bed, dispersion energetics, etc.

*2.6. Thermo-Gravimetric Analysis of TCD Carbon*

Quantification of the TCD carbon upon the nascent catalyst can be made by differential oxidation. Given the different nanostructures of the TCD carbon and the initial carbon catalyst, their oxidative reactivity may be sufficiently different as to render these $sp^2$ phase fractions separable by partial oxidation.

Recall that the nanostructures were inferred to be different based on the following observations:

1. Different TCD rates at short versus long reaction times—wherein the nascent carbon catalyst controlled TCD initially; whereas at longer durations the asymptotic rate was governed by the TCD carbon, and;
2. TEM images of the carbon catalyst post-TCD show that the partially covered original carbon was found otherwise unchanged, whereas the TCD carbon was identified as having a coral-like structure consisting of folded sheets in filamentary form, somewhat similar to wrinkled lamellae extending outward from the nascent carbon black particles. (For the R250 and sugar char with methane.)

Therein, a slow temperature ramp in the air can exploit the different reactivities of the two carbon forms, and thermogravimetric analysis can resolve the two forms by mass loss. If the respective oxidation ranges are blended, then separation can be achieved by differentiation, as commonly applied to TGA spectra. Advantages of the derivative are that it resolves oxidation stages by rates, yielding identifiable (thermo-gravimetric) spectral peaks by which to identify stages and determine when the rate of oxidation goes to zero for a particular spectral peak, the two limits being associated with the onset and conclusion of oxidation of a particular form/phase of carbon. If spectral peaks overlap, curve fitting and deconvolution effectively resolve these limits. Additionally, the area under a spectral peak integrates the rate, effectively yielding the mass associated with the particular carbon phase. Finally, the position of the peaks marks the maximum oxidation rate for the respective carbon forms.

These concepts are illustrated for the R250 catalyst extracted from the packed-bed reactor after TCD in methane at 900 °C in Figure 10. Identification of the phases was made by comparison to corresponding TGA spectra and derivative curves for reference carbon materials, e.g., the nascent carbon catalyst. As there is no "standard TCD carbon" material, a carbon form with similar morphology and a high edge to basal carbon structure was chosen—commercial graphene. With its submicron platelet size and wrinkled morphology, it bears resemblance to TCD carbon.

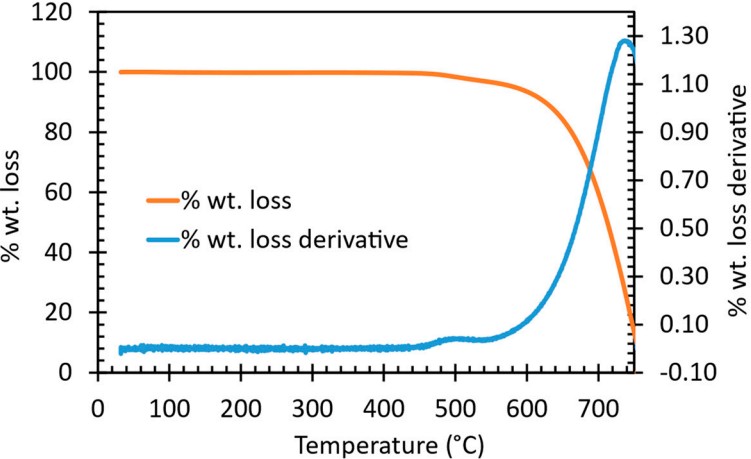

**Figure 10.** The TGA spectrum and its derivative for the post-TCD R250 carbon.

A small peak identified at 500 °C was assigned to the TCD carbon coal with a fractional contribution to the total mass of ~5%. As Figure 9 shows, the carbon outgrowth exhibited a non-uniform structure with graphene segments along with twists and folds in these, increasing the reactivity of exposed basal and edge site carbons.

### 2.7. Gauging Regeneration—Via Partial Oxidation

To gauge the efficacy of regeneration by partial oxidation, comparative TCD runs were performed using nascent R250 and its partially oxidized form (~58% burn-off level) using synthetic natural gas (SNG). As stated previously, the purpose of regeneration is to recreate active sites through partial oxidation. Normally, RGN would be applied to TCD

carbon after a significant loss of activity. But for reference to a well-defined nanostructure and uniform carbon morphology (rather than an admixture of TCD carbon upon another carbon), partial oxidation was performed on the nascent R250 in situ. Figure 11 shows comparative TEM images for the two carbons, illustrating the differences in particle morphology. Specifically, the partial oxidation performed at 500 °C in air produces hollow shells, a result of preferential interior oxidation of low-structured carbon. Figure 12 shows the TCD runs for each carbon.

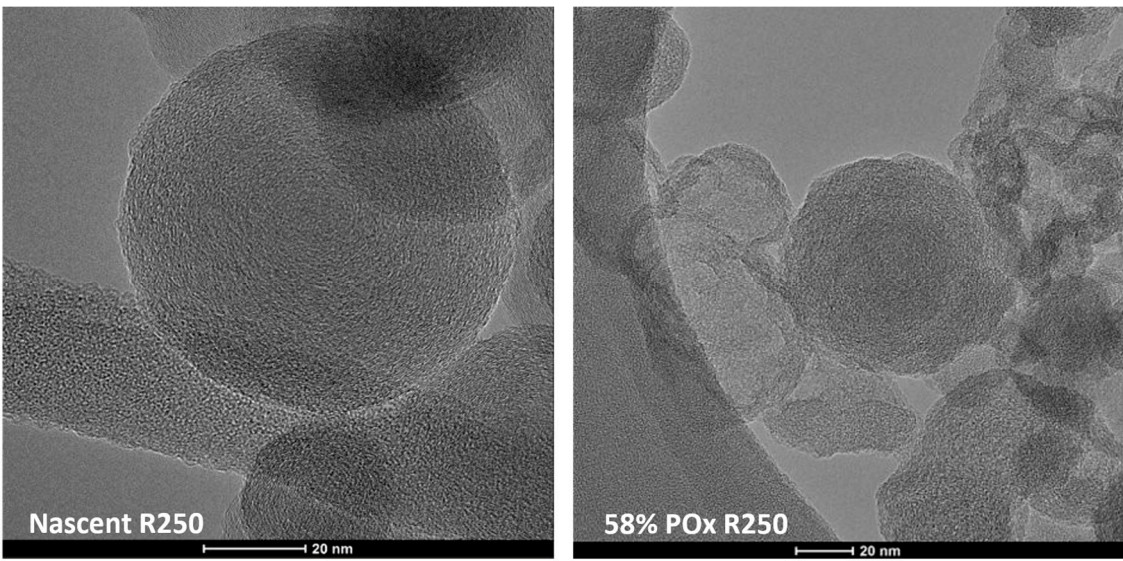

**Figure 11.** TEM images contrasting the nanostructure of nascent and partially oxidized R250.

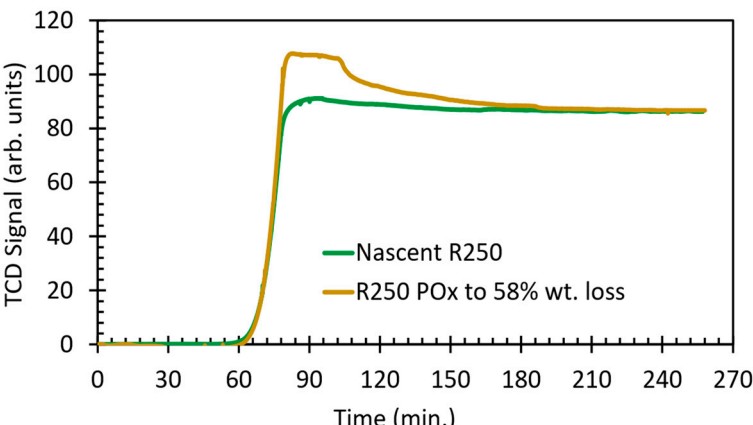

**Figure 12.** TCD runs utilizing nascent R250 and its partially oxidized form, with SNG at 800 °C.

As Figure 12 shows, the onset temperatures are equivalent, as are the activation energies, though the partially oxidized form does, as expected, exhibit a higher peak activity. Interestingly, its activity remains larger than the nascent R250, though eventually the two carbons have equivalent conversion—again a reflection of the TCD carbon governing activity with negligible contribution from the now buried; original carbon catalyst.

Given the previously observed reactivity differences between TCD carbons and the lack of such differences as observed in Figure 12 above, TEM was performed on the recovered, partially oxidized R250 carbon post-TCD. Figure 13 presents a TEM comparison between the partially oxidized R250, pre- and post-TCD. As observed, the TCD carbon was added to the oxidized carbon shells, increasing their apparent thickness and overall particle diameter (illustrated by the arrows in Figure 13). The radial continuity of the nanostructure is high; the boundary between the nascent particle and the added TCD carbon is effectively unrecognizable. Effectively, the TCD carbon appears to have templated

upon the partially oxidized carbon. Here too, the TCD carbon nanostructure appears to depend on the nanostructure of the original carbon catalyst. In addition to serving as a comparison for nanostructure-based activity, these measurements provide a baseline for gauging the effect of partial oxidation (RGN), which presumably should increase the active sites (and corresponding change in nanostructure).

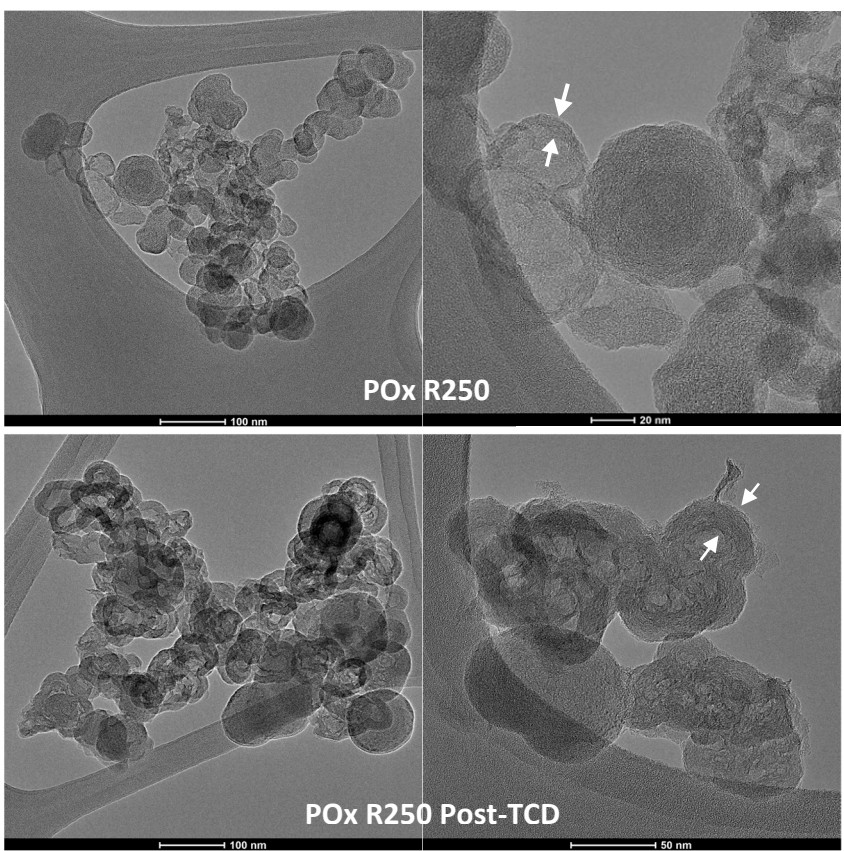

**Figure 13.** TEM comparison between partially oxidized R250, pre- and post-TCD. Note the different magnification scales between the images.

### 2.8. Activation Energy Analysis

Activation energies were extracted from the TCD rates, analogous to the leading-edge analysis in thermal desorption spectroscopy [36] or the single ramp rate method used in TGA analysis [37,38] for R250 and its partially oxidized form (25%) with methane. Corresponding TCD results are shown in Figure 14. With the approximation that there is little deposited carbon on the initial catalyst during the ramp to operating temperature (hence the initial catalyst yet governs the TCD rate once at temperature), rates can be extracted by analyzing the initial TCD conversion with time. Following an Arrhenius analysis for the activation energy, the TCD activation energy with nascent R250 with methane was 281 kJ/mol but decreased to 133 kJ/mol for the partially oxidized R250. This dramatic decrease confirms the regeneration/re-creation of active sites by partial oxidation and, moreover, that sites formed by oxidation are active towards deposition. Notably, the earlier TCD onset temperature (see Figure 13) is consistent with the lower activation energy value for the partially oxidized carbon. It is also interesting to note that these activation energies span the range reported in the literature for carbon black catalysts [13,19,39].

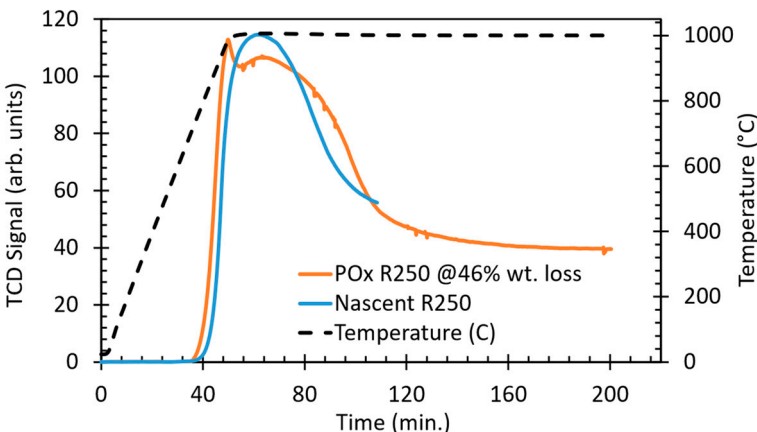

**Figure 14.** TCD plots correspond to the conversion of methane into carbon and hydrogen.

By XPS analysis for chemisorbed oxygen following activated chemisorption, the number of active sites increased by a factor of 7 upon partial oxidation to 30% mass loss. While the number of active sites increased, their intrinsic reactivity was different—as evident by the change in activation energy. Analysis of the carbon-oxygen bonding reveals a shift from a -C=O/-C-O bond ratio greater than 1 for the nascent form to a ratio less than 1 upon partial oxidation, a signature of the oxidation mechanism.

### 2.9. Connecting Regeneration with Nanostructure and Soot Oxidation

The connection between regeneration rate and active sites pertains not only to RGN in connection with TCD but more broadly to combustion-generated soot. Engine and combustor soot nanostructure depends upon the formation conditions [40,41] and has been shown to strongly depend upon the local equivalence ratio [42,43]. While oxidation rates for model soots have been shown to correlate with nanostructure metrics of fringe length, tortuosity, and separation distance [44,45], the root foundation of the correlation, namely active sites, is not measured. Nanostructure can serve as a viable surrogate with multiple structural metrics by which to understand oxidation rates, but establishing a proven connection between nanostructure and active sites would provide a solid foundation for future use of nanostructure for regeneration rate interpretation.

The basis for the proposed relationship between nanostructure and active sites is straightforward: shorter lamellae possess an increasing number of edge sites relative to basal sites. Hence, active sites should scale inversely with lamellae length (or distribution), adopting a circular representation for lamellae described by fringe length. This simplistic interpretation neglects other potential reaction sites such as vacancies, step edges, stacked edges, diradical sites, etc. It also presumes that basal sites are inactive while not contributing any secondary role, such as the adsorption of reactive species followed by lateral diffusion/migration to active sites. Still, as a first-order model, the geometric relationship does comport with the HACA mechanism. However, not all edge sites may be (or become) active sites, in which case a proportionality factor would be necessary. An additional consideration, as discussed subsequently (future work), is that sites active for TCD may not correspond fully to sites active in RGN.

Yet initial observations suggest that such a simple relationship between nanostructure and active sites may be more complex than a geometric one based on lamellar size. Shown in Figure 15 are contrasting behaviors of regeneration by oxidation upon nanostructure.

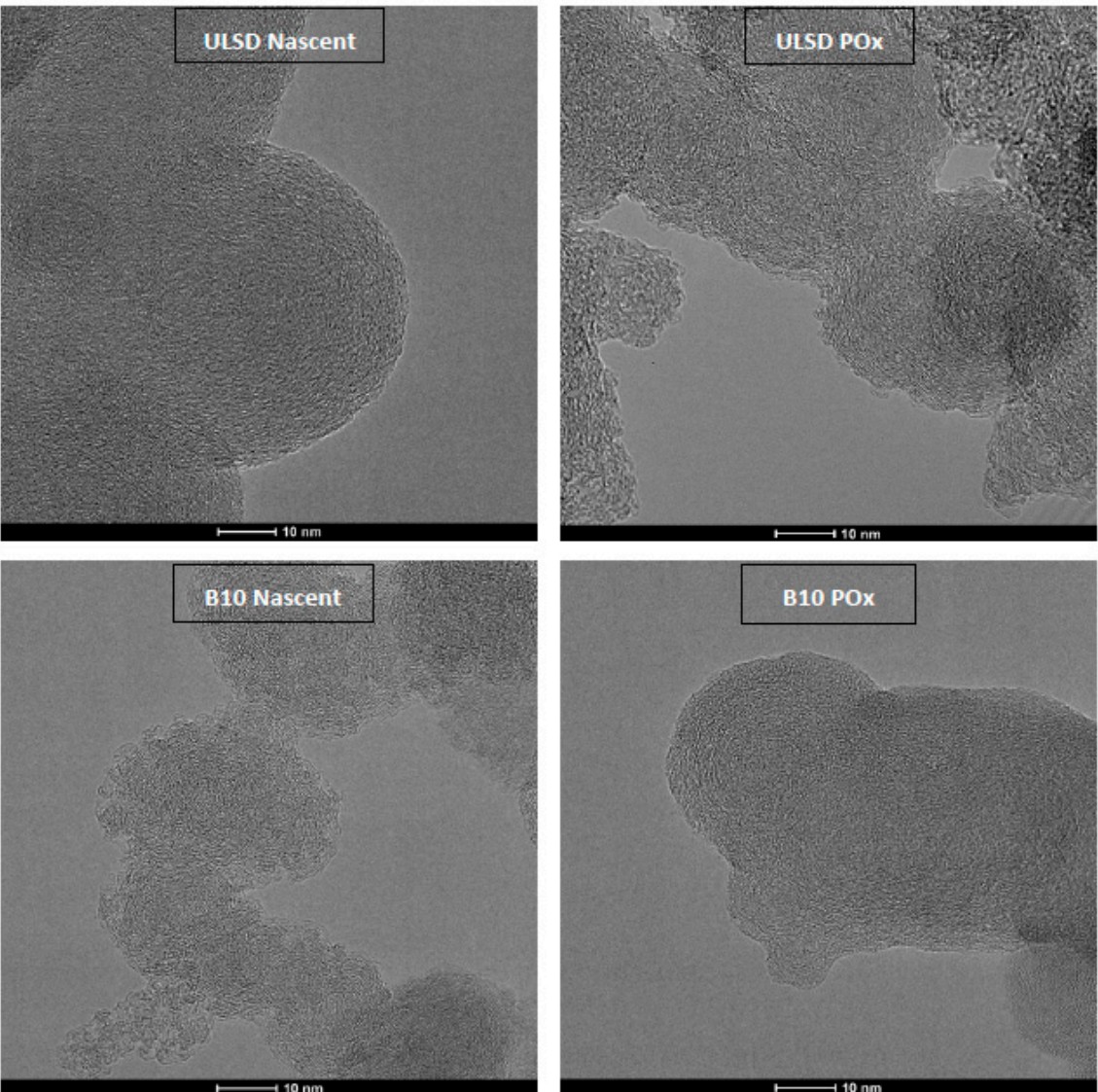

**Figure 15.** TEM images (nascent and post-oxidation) of ULSD and B10 soots produced by the corresponding fuels in a diesel engine.

The top row shows nascent and partially oxidized ULSD soot. Focusing on the particle surface, the nascent form exhibits a smooth profile with no evident disruption of layered lamellae. In contrast, the partially oxidized form has a rougher appearance. Closer inspection reveals numerous peripheral gaps and tightly curved lamellae, as marked by the arrows. Both are signs of lamellae liberation and/or breakup by the removal of cross links, bridging connections, etc. Unpinned and with reactive edges, lamellae curl to join dangling bonds.

Oppositely, the nascent B10 soot exhibits significant lamellar curvature, as reported previously [44], due to partial premixing. Along the particle perimeter, numerous shells, curved lamellae, and irregularities appear, as marked by the arrows. Upon partial oxidation, this structural mix is reduced to an amorphous layer, nanometers in thickness, that surrounds the inner particles. The high bond strain associated with the high curvature enables oxidative attack of basal sites and lamellae breakup. Active sites for both carbons increased relative to the nascent carbon upon partial oxidation to ~10 at.%, essentially the same despite the very different peripheral nanostructures evident in Figure 15. While the increase in active sites can be understood in terms of the change in nanostructure, a simple relationship between lamellae length and active sites may be inadequate. Moreover, both

nascent carbons challenge fringe analysis, thus motivating other techniques such as Raman analysis for structural characterization and possible relationship to active sites.

### 3. Experimental Approach

#### 3.1. Packed Bed

The principal advantage of the packed bed is that its scale enables monitoring reaction kinetics by reaction products with standard instrumentation. Product analyses furnish reaction metrics of yield, product selectivity, and catalyst stability. Parameters of interest include catalyst activity and stability. The challenge in determining each of these metrics lies in the intermediate regime, where both the initial carbon catalyst and evolving TCD carbon both contribute to the overall reaction/conversion. Depending on the relative reactivity of the two different carbons, varied time behaviors can be observed for the reaction rate and conversion yield with time. At longer durations, it can be safely assumed that only the deposited carbon is contributing to the observed activity, due to the coverage of the initial catalyst and the asymptotic behavior of the observed reaction rate.

#### 3.2. Reactor

Using a commercial microreactor system (AMI-300 chemisorption analyzer—University Park, PA 16802, USA), the starting carbon catalysts are placed in a U-shaped quartz tube held in a vertical reactor. Mass flow controllers regulate the flow consisting of a mixture of inert carrier and hydrocarbon gas, or mixture for TCD or inert plus oxidizer, such as $CO_2$, $H_2O$, or $O_2$ for regeneration. Total gas flow was kept at 25 sccm, and reactive-to-inert flows varied depending on the experiment. The initial catalyst mass was ~15 to 20 mg. Oxidation levels are gauged initially by matching in situ regeneration conditions to corresponding thermogravimetric analysis (TGA) data for the same carbons and regeneration (RGN) conditions. A thermal conductivity detector is being used for monitoring reaction progress in TCD and regeneration (RGN). Space velocities were ~14 $s^{-1}/g_{cat}$.

#### 3.3. HRTEM

High-resolution transmission electron microscopy (HRTEM) has been carried out using the 200 keV FEG source of an FEI Talos F200X (University Park, PA 16802, USA) with a resolution of 0.12 Å. Samples were dispersed and sonicated in methanol before being dropped onto 300-mesh C/Cu-lacey TEM grids.

#### 3.4. EDS

Energy dispersive X-ray spectroscopy (EDS) for elemental analysis and mapping has been performed in the TEM (FEI Talos) in scanning transmission electron microscopy (STEM) mode with a sample holder designed to provide a low background signal for EDS (refer to Figures S1 and S2 in the Supplementary Material). STEM mode offers a high spatial resolution on the order of the minimum probe size, which is 1.6Å. The Talos uses the Super X-EDS system, which is comprised of four silicon drift detectors that produce very large X-ray counts to allow for a better signal-to-noise ratio. The high counts from the large area of the detectors also provide for very low detection limits of typically < 1 atomic percent (at.%), depending on collection parameters.

#### 3.5. XPS

X-ray photoelectron spectroscopy (XPS) measurements were performed using a Physical Electronics VersaProbe II instrument equipped with a monochromatic Al Kα X-ray source (hν = 1.4867 keV) and a concentric hemispherical analyzer. Charge neutralization was performed using both low-energy electrons (<5 eV) and argon ions. The BE axis was calibrated using sputter-cleaned Cu foil (Cu 2p3/2 = 932.7 eV, Cu 2p3/2 = 75.1 eV). Peaks were charge-referenced to the C-C band in the carbon 1s (C1s) spectra at 284.5 eV. Measurements were made at a takeoff angle of 45° with respect to the sample surface plane. This resulted in a typical sampling depth of 3–6 nm (95% of the signal originated from this

depth or shallower). Quantification was carried out using instrumental relative sensitivity factors (RSFs) that account for the X-ray cross-section and inelastic mean free path of the electrons. Chemisorbed oxygen is equated to active sites, as during the initial ramp to reaction temperature, these groups decompose, producing CO and $CO_2$, leaving behind radical sites arising from the carbon loss [46–48].

### 3.6. Activated Chemisorption

To measure active sites, carbons were first heated to 500 °C under an inert atmosphere before cooling to 300 °C, followed by exposure to air. After $\frac{1}{2}$ h. exposure, samples were cooled to ambient for subsequent XPS analysis for chemisorbed oxygen, resolved by bonding to the carbon, and quantified as atomic oxygen percent.

### 3.7. Carbon Catalysts

Carbon blacks were obtained from commercial producers: Ketjenblack from AkzoNobel and R250 from Cabot Corp. The graphitized R250 was formed by heating the nascent R250 to 3000 °C using a graphitization furnace. The sugar char was produced by the carbonization of sucrose at 500 °C.

### 4. Conclusions

The TCD carbon nanostructure poses a serious challenge to quantification by fringe analysis, given its varied forms and apparent dependence upon the initial carbon catalyst. Both sugar char and R250 supported carbon coral from methane, while SNG led to new particle growth and uniform templating upon partially oxidized R250. Therein, alternative metrics for structure may be warranted, such as Raman spectroscopy.

The connections between the carbon catalyst structure and active sites in TCD are presently qualitative. To date, the focus has largely been on the initial catalyst rather than the TCD carbon itself acting as a catalyst. The structure of this deposited carbon (catalyst) depends on the hydrocarbon feed with pure methane, leading to coral-like deposits. If structure can be correlated to active sites, a surrogate metric will be established by which to gauge carbon structure for reactivity under TCD and regeneration conditions. Structure, specifically nanostructure, provides a more straightforward method of assessing carbon reactivity based on structure than active site measurement, the latter being very dependent upon sample preparation and chemisorption procedures. challenging the general prescription that a more disordered structure is more active; an amorphous sugar char was less active than a conductive carbon black. Similarly, an unstructured coating on graphitized carbon black proved less active than the nascent form with recognizable lamellae.

Significantly, regeneration increases carbon reactivity, as manifested by an earlier onset temperature, a faster initial rate, and higher conversion. More importantly, this observation confirms that active sites created by oxidation are active in TCD. Ideally, regeneration would be conducted using $CO_2$ in a swing-bed system in conjunction with TCD. The CO formed by the regeneration (gasification) reaction could be combined with TCD-produced $H_2$ for Fischer-Tropsch synthesis of liquid fuels. Coupled to renewable energy, the hydrogen generated by TCD accomplishes renewable energy storage, namely the conversion of electrical energy into chemical bond energy. Finally, hydrogen addition to natural gas for power generation is underway in Europe and the U.S. as a means of decarbonizing this energy sector.

**Supplementary Materials:** The following supporting information can be downloaded at: https://www.mdpi.com/article/10.3390/catal13101382/s1; Figure S1: EDS scan, elemental map and HAADF image for the nascent B10 soot.; Figure S2: EDS scan, elemental map and HAADF image for the partially oxidized (50%) B10 soot.

**Author Contributions:** Conceptualization, R.V.W. and M.M.N.; methodology, R.V.W. and M.M.N.; software, M.M.N.; validation, R.V.W. and M.M.N.; formal analysis, R.V.W. and M.M.N.; investigation, R.V.W. and M.M.N.; data curation, R.V.W. and M.M.N.; writing—original draft preparation, R.V.W.; writing—review and editing, M.M.N.; visualization, M.M.N.; supervision, R.V.W.; project administration, R.V.W.; funding acquisition, R.V.W. All authors have read and agreed to the published version of the manuscript.

**Funding:** This work was supported by the U.S. Department of Energy, Office of Science, Office of Basic Energy Sciences, Gas Phase Chemical Physics program, under Award number DE-SC0021059.

**Data Availability Statement:** The data presented in this study are available on request from the corresponding author.

**Conflicts of Interest:** The authors declare no conflict of interest.

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
