# Peer review of "Thermo-Catalytic Decomposition Comparisons: Carbon Catalyst Structure, Hydrocarbon Feed and Regeneration"

_catalysts, doi:10.3390/catal13101382_

Round 1

Reviewer 1 Report

I have been reviewed the manuscript titled "Thermo-catalytic Decomposition Comparisons: Carbon Catalyst Structure, Hydrocarbon Feed and Regeneration".

Introduction:

In the first paragraph from line 35 upto line 38 it’s not arranged text and not clear.

You should clarify what it means: TCD, WGS, SNG, TGA, RGN, and CNT, for the first time in the draft.

Previous studies and examples show some prepared catalysts with their catalytic results for hydrogen/ regenerations are missing. 

Also, the novelty of the work is not clear, what is this work bringing new in this interesting area.

Results:

In figure 2 and 3 you need to shows the clear different on the pictures to be clearer (i.e. arrows),

the XPS analysis in table or as figure to improve the text.

When you start to present your results you show a “3.1. Nanostructure Comparisons: Sugar Char Versus Carbon Black” then you started to talk about the 3.2. Kinetic Rates with other different material “graphitized R250, (G-R250)”??, what is the correlation?

In the part at line 254-256: “Indeed, the earlier onset temperature suggests that the activation energy for the G-R250 form is significantly lower than that for the R250” calculations are required.

Introduce some information about “pre, post-TCD”

In lines 292 – 294: you said “Several prior publications presenting TEM images of the G-R250 form showed clean basal planes with negligible carbon debris, as illustrated in Figure 3”, is this figure taken from a reference not from your current work?

Introduce the Energy dispersive X-ray spectroscopy (EDS)  in the main draft. 

It is advisable to consider rewriting the manuscript to improve the overview of the work.

I have been reviewed the manuscript titled "Thermo-catalytic Decomposition Comparisons: Carbon Catalyst Structure, Hydrocarbon Feed and Regeneration".

Introduction:

In the first paragraph from line 35 upto line 38 it’s not arranged text and not clear.

You should clarify what it means: TCD, WGS, SNG, TGA, RGN, and CNT, for the first time in the draft.

Previous studies and examples show some prepared catalysts with their catalytic results for hydrogen/ regenerations are missing. 

Also, the novelty of the work is not clear, what is this work bringing new in this interesting area.

Results:

In figure 2 and 3 you need to shows the clear different on the pictures to be clearer (i.e. arrows),

the XPS analysis in table or as figure to improve the text.

When you start to present your results you show a “3.1. Nanostructure Comparisons: Sugar Char Versus Carbon Black” then you started to talk about the 3.2. Kinetic Rates with other different material “graphitized R250, (G-R250)”??, what is the correlation?

In the part at line 254-256: “Indeed, the earlier onset temperature suggests that the activation energy for the G-R250 form is significantly lower than that for the R250” calculations are required.

Introduce some information about “pre, post-TCD”

In lines 292 – 294: you said “Several prior publications presenting TEM images of the G-R250 form showed clean basal planes with negligible carbon debris, as illustrated in Figure 3”, is this figure taken from a reference not from your current work?

Introduce the Energy dispersive X-ray spectroscopy (EDS)  in the main draft. 

It is advisable to consider rewriting the manuscript to improve the overview of the work.

Reviewer 2 Report

I recommend rejecting the manuscript in its current form due to numerous issues that compromise its scientific rigor and clarity. The manuscript focuses on Thermo-catalytic Decomposition (TCD) activity and its relationship with different types of carbon catalysts. While the subject matter is of scientific relevance, there are multiple critical issues that undermine the quality and credibility of the work.

1The manuscript has several structural issues that disrupt its flow and readability. Lines 36 and 37 consist of a single sentence that forms an entire paragraph, which is not in accordance with standard academic writing. Additionally, Figure 2 contains disorganized post-TCD text, and Figure 3 lacks essential annotations for lattice spacing. The text "post-TCD" on line 270 is also inexplicably bolded, adding to the confusion.

2 The abstract mentions the use of X-ray Photoelectron Spectroscopy (XPS) for characterizing active sites. However, this characterization method is conspicuously absent in the main body of the manuscript, creating a gap between the abstract and the actual content.

3 While the manuscript heavily relies on TEM images for characterizing Carbon Catalyst Structure, it lacks basic but crucial characterizations like XRD and Raman spectroscopy. These are fundamental techniques for a comprehensive understanding of catalysts and their properties.

The manuscript is clearly written and the English language used is understandable. There are no major issues with the quality of the English language that would hinder comprehension of the scientific content.

Reviewer 3 Report

G S F C University,

3rd September 2023.

Manuscript ID: Catalysts-2596689

Title: Thermo-catalytic decomposition comparisons: Carbon catalyst structure, hydrocarbon feed and regeneration

Authors: Mpila Nkiawete, and Randy Vander Wal*

Summary: The authors, namely, Nkiawete and Vander Wal made an attempt to study the vital reaction, i.e., the thermocatalytic decomposition of natural gas using carbocatalysts such as sugar char, carbon blacks and graphitized carbon black.  The effect of the structure of the carbocatalyst, hydrocarbon feed and the regeneration property on the thermocatalytic decomposition of hydrocarbon is examined using thermal analysis (TG-DTA), gas chromatography using thermal conductivity detector, HR-TEM, and EDAX analysis. Useful deductions with implications on the production of hydrogen fuel from natural gas using carbocatalysts via thermocatalytic decomposition were made. Owing to the originality and usefulness of the results the work of Nkiawate and Vander Wal is recommended for publication in Catalysts after major revision.

Major issues:

English language need to be significantly improved throughout the manuscript for making the content understandable.

Lines (L)  196-198: The text referring to the onset temperature while explain the figure 1 is unclear. Please explain how does the temperatures of 650 and 750 °C correspond to the onset temperature of the char and the carbon black.  The term “onset temperature” need to be first of all clearly defined.

Minor issues:

In addition to the above, the following minor but important issues need to be addressed for enhancing the quality of the manuscript.

Line (L) number

Revision

299

Caption of figure 6: Replace G-250 with G-R250.  Such correction need to be made throughout th3e paper wherever applicable.  Also add “Post TCD” before “R250”

261

Replace “G-250” with “G-R250”

262

Delete “yet”

411

Replace “R250 carbon post-TCD” with “Post-TCD R250 carbon”

17

Add “for the TCD of” before “nascent”

17

Replace “carbon” with “carbons”

119

Add “its” before “dependence”

23

Replace “evaluated for its potential” with “carried out”

9-11

This text is unclear and the thoughts of the authors are not clearly expressed. Rather the text could be rewritten as follows:

Response to the thermocatalytic decomposition (TCD) depends on the initial carbocatalyst structure.  However, further transition in the carbon structure depends on the carbon material (structure and composition) originating from the TCD process.

17

Add “for the partial oxidation as well as the TCD process” after “Activation energies”

17

Provide the magnitude of the activation energies for the partial oxidation as well as the TCD processes carried out on the nascent and the oxidized carbon structures.

18

Replace “connections” with “correlation”

27

Replace “of” with “for”

28

Delete “use”

29

Add “generating” before “electricity”

29

Delete “each”

29

Add “gas” before “emission”

30

Replace “CO2” with “CO2”; apply this change throughout the paper.  Subscripts and superscripts to be written as per the convention.

30-31

Replace “It also” with “Hydrogen”

31

Replace “storage vehicle for renewable energy” with “sustainable renewable energy resource.”

33

Replace “for” with “from”

33

Replace “varied” with “various”

35

Replace “The majority of hydrogen today” with “Today, majority of the hydrogen”

36

Delete “Numbers”

40

Replace “impose” with “result in”

43

Add “water gas shift” before “WGS”

43

Add “with reaction” after “WGS”

35

Replace “steam methane reforming (SMR)” with “methane steam reforming (MSR)”

43

Replace “SRM” with “MSR”

48

Replace “CO2” with “CO2

175

Replace “C” with “°C”’ apply this change throughout the paper wherever applicable

586-589

Reference number “11” is repeated twice

579-584

Replace “CO2” with “CO2”; this change need to be applied throughout the paper

133

“CO2, H2O, O2” to be written as “CO2, H2O, O2

137

“gr-1s-1” to be written as “s-1”; What is the unit of space velocity?

139

Replace “microscopy” with “microscopic analysis”

589

The reference is incomplete; doi of the reference can be provided

606

The reference is incomplete; page numbers, year and volume were missing

619

“H2” to be written as “H2

638

Reference 35 is incomplete; ISBN or page number and volume and the publishers name were missing

66

Replace “voiding” with “avoiding”

79

What is TCD stability” The sentence is unclear.

81-82

The sentence is unclear. What does “Long term” mean?

86

How can the gas feed have crystallite size? The sentence is unclear.

102-103

The sentence, “There lies the importance of evaluate TCD with TCD carbon” is unclear.

104-107

The sentence is too long and unclear

112-113

Replace “upon that of the” with “compared to the”

113

Replace “for” with “of”

113

Replace “active site formation with “active sites”

115

Replace “given their” with “owing to their”

165

Replace “are” with “the”

167

Add “atmosphere” after “inert”

137

To which instrument does the thermal conductivity detector correspond to? If a gas chromatograph were to be used for the analysis, the specification need to be provided.

G S F C University,

3rd September 2023.

Manuscript ID: Catalysts-2596689

Title: Thermo-catalytic decomposition comparisons: Carbon catalyst structure, hydrocarbon feed and regeneration

Authors: Mpila Nkiawete, and Randy Vander Wal*

Summary: The authors, namely, Nkiawete and Vander Wal made an attempt to study the vital reaction, i.e., the thermocatalytic decomposition of natural gas using carbocatalysts such as sugar char, carbon blacks and graphitized carbon black.  The effect of the structure of the carbocatalyst, hydrocarbon feed and the regeneration property on the thermocatalytic decomposition of hydrocarbon is examined using thermal analysis (TG-DTA), gas chromatography using thermal conductivity detector, HR-TEM, and EDAX analysis. Useful deductions with implications on the production of hydrogen fuel from natural gas using carbocatalysts via thermocatalytic decomposition were made. Owing to the originality and usefulness of the results the work of Nkiawate and Vander Wal is recommended for publication in Catalysts after major revision.

Major issues:

English language need to be significantly improved throughout the manuscript for making the content understandable.

Lines (L)  196-198: The text referring to the onset temperature while explain the figure 1 is unclear. Please explain how does the temperatures of 650 and 750 °C correspond to the onset temperature of the char and the carbon black.  The term “onset temperature” need to be first of all clearly defined.

Minor issues:

In addition to the above, the following minor but important issues need to be addressed for enhancing the quality of the manuscript.

Line (L) number

Revision

299

Caption of figure 6: Replace G-250 with G-R250.  Such correction need to be made throughout th3e paper wherever applicable.  Also add “Post TCD” before “R250”

261

Replace “G-250” with “G-R250”

262

Delete “yet”

411

Replace “R250 carbon post-TCD” with “Post-TCD R250 carbon”

17

Add “for the TCD of” before “nascent”

17

Replace “carbon” with “carbons”

119

Add “its” before “dependence”

23

Replace “evaluated for its potential” with “carried out”

9-11

This text is unclear and the thoughts of the authors are not clearly expressed. Rather the text could be rewritten as follows:

Response to the thermocatalytic decomposition (TCD) depends on the initial carbocatalyst structure.  However, further transition in the carbon structure depends on the carbon material (structure and composition) originating from the TCD process.

17

Add “for the partial oxidation as well as the TCD process” after “Activation energies”

17

Provide the magnitude of the activation energies for the partial oxidation as well as the TCD processes carried out on the nascent and the oxidized carbon structures.

18

Replace “connections” with “correlation”

27

Replace “of” with “for”

28

Delete “use”

29

Add “generating” before “electricity”

29

Delete “each”

29

Add “gas” before “emission”

30

Replace “CO2” with “CO2”; apply this change throughout the paper.  Subscripts and superscripts to be written as per the convention.

30-31

Replace “It also” with “Hydrogen”

31

Replace “storage vehicle for renewable energy” with “sustainable renewable energy resource.”

33

Replace “for” with “from”

33

Replace “varied” with “various”

35

Replace “The majority of hydrogen today” with “Today, majority of the hydrogen”

36

Delete “Numbers”

40

Replace “impose” with “result in”

43

Add “water gas shift” before “WGS”

43

Add “with reaction” after “WGS”

35

Replace “steam methane reforming (SMR)” with “methane steam reforming (MSR)”

43

Replace “SRM” with “MSR”

48

Replace “CO2” with “CO2

175

Replace “C” with “°C”’ apply this change throughout the paper wherever applicable

586-589

Reference number “11” is repeated twice

579-584

Replace “CO2” with “CO2”; this change need to be applied throughout the paper

133

“CO2, H2O, O2” to be written as “CO2, H2O, O2

137

“gr-1s-1” to be written as “s-1”; What is the unit of space velocity?

139

Replace “microscopy” with “microscopic analysis”

589

The reference is incomplete; doi of the reference can be provided

606

The reference is incomplete; page numbers, year and volume were missing

619

“H2” to be written as “H2

638

Reference 35 is incomplete; ISBN or page number and volume and the publishers name were missing

66

Replace “voiding” with “avoiding”

79

What is TCD stability” The sentence is unclear.

81-82

The sentence is unclear. What does “Long term” mean?

86

How can the gas feed have crystallite size? The sentence is unclear.

102-103

The sentence, “There lies the importance of evaluate TCD with TCD carbon” is unclear.

104-107

The sentence is too long and unclear

112-113

Replace “upon that of the” with “compared to the”

113

Replace “for” with “of”

113

Replace “active site formation with “active sites”

115

Replace “given their” with “owing to their”

165

Replace “are” with “the”

167

Add “atmosphere” after “inert”

137

To which instrument does the thermal conductivity detector correspond to? If a gas chromatograph were to be used for the analysis, the specification need to be provided.

Reviewer 4 Report

The authors aim at evaluating the thermo-catalytic decomposition reaction using carbon as catalysts evaluating the role of the carbon formed upon TCD on the rate of reaction, by analyzing TEM images post-TCD. The results are more qualitative rather than quantitative, but they provide interesting and important insights into a technology that will be valuable in the future. The paper is well-written, although some passages may be trickier to read. I have a few suggestions for the authors that may improve the paper:

1. In general, the introduction is well-written. Nevertheless, in some sections there are some issues that make it more difficult to follow, such as: 

-In line 36, the word "numbers" appears alone. Does it mean something? Feels like it is misplaced

-TCD is introduced in line 37 but the meaning of the acronym is only given in line 45.

- WGS (Lines 43 and 46) is not given anywhere in the manuscript

- the unity of the space velocity in line 137 seems incorrect

2. The conditions used during TCD could be further detailed  (Flow, ratio between inert gas and hydrocarbons, catalysts mass...). 

3. Other information regarding the carbon samples utilized could be useful (carbon content, ashes, etc)

4.The authors mention that the carbons utilized are non porous, however they do not provide the surface area analysis. I believe adding surface area analysis could be useful. 

5. The authors use XPS to quantify oxygen in the samples, but the spectra is not provided. I believe it could be also useful to provide the XPS spectra of the samples. 

6. How long can this system operate before the carbon catalyst requiring regeneration?

7. How do the results reported here compare to the literature? 

The english language is fine.

Round 2

Reviewer 1 Report

The draft doesn't require any further changes or adjustments.

Reviewer 2 Report

After reviewing the manuscript, I believe it possesses merit and is worthy of publication. I recommend that this manuscript be accepted for publication.